# Novel Blue-Wavelength-Blocking Contact Lens with Er^3+^/TiO_2_ NPs: Manufacture and Characterization

**DOI:** 10.3390/nano11092190

**Published:** 2021-08-26

**Authors:** Lina Mohammed Shaker, Ahmed Alamiery, Mohd Takriff, Wan Nor Roslam Wan Isahak

**Affiliations:** 1Department of Chemical and Process Engineering, Faculty of Engineering and Built Environment, Universiti Kebangsaan Malaysia (UKM), Bangi 43000, Selangor, Malaysia; wannorroslam@ukm.edu.my (L.M.S.); sobritakriff@ukm.edu.my (M.T.); p109960@siswa.ukm.edu.my (W.N.R.W.I.); 2Energy and Renewable Energies Technology Center, University of Technology, Baghdad 10001, Iraq; 3Chemical and Water Desalination Engineering Program, Collage of Engineering, University of Sharjah, Sharjah 26666, United Arab Emirates

**Keywords:** titanium dioxide, erbium ions, high surface area, impregnation method, optical characteristics

## Abstract

Thermally stable titanium dioxide nanoparticles (TiO_2_ NPs) doped with erbium ions (Er^3+^) are characterized by uniformity, low excitation energy, and high surface area. The impregnation methodology was used to enhance the optical properties of TiO_2_ NPs impregnated with various Er^3+^ ion contents. The synthesized Er^3+^/TiO_2_ samples were characterized by energy dispersive X-ray (EDX), metal mapping, UV–Visible spectrum, field emission scanning electron microscopy (FESEM), and X-ray diffraction (XRD). The Er^3+^ ions, per our findings, were well-distributed on the TiO_2_ surface of the anatase phase and there was an insignificant difference in particle size, but there was no change in the particle shapes of the Er^3+^/TiO_2_ NPs structure. The maximum band gap degradation occurred with 1.8 wt % of Er^3+^/TiO_2_, where the energy gap degraded from 3.13 to 2.63 eV for intrinsic TiO_2_. The synthesized Er^3+^/TiO_2_ samples possess predominantly finely dispersed erbium ion species on the surface. Er^3+^ ions agglomeration on the surface increased with increasing ions in each sample. We found that 0.6 wt/vol % of Er^+3^/TiO_2_ is the best optical coating and produced satisfying results in terms of blocking the transmittance of blue wavelength without reducing the image quality.

## 1. Introduction

Recently, various methods have been studied and applied for modifying mineral surfaces [1,2]. Such methods enhance the activity of titanium dioxide (TiO_2_) for different applications [3], for example, as a photocatalyst for the photocatalytic degradation (PCD) of pollutants [4], both organic and inorganic [5,6], and for sterilizing wastewater [7]. A photocatalytic antimicrobial film was coated on a polymeric contact lens surface to fabricate an anti-contamination contact lens and other applications [8]. Hybrid poly(methyl methacrylate-dioxide) systems have been widely reported, focusing mostly on PMMA-SiO_2_ [9], PMMA-TiO_2_ [10], and PMMA-ZrO_2_ [11]; these nanocomposites are used in the fabrication of optical and biomedical devices such as contact lenses [11]. Nanocomposite polymers have attracted significant attention in optical applications because of their enhanced optical performance. Recently, nanoparticles (NPs) such as TiO_2_, ZnO, ZnS, ZrO_2_, and so forth, have been exploited to obtain a high refractive index (*n*) and low dispersion nanocomposite [12,13]. Switchable wettability materials such as TiO_2_-based polymers are considered good photocatalytic coating candidates able to block the undesired ultraviolet (UV) spectrum [14].

The TiO_2_ photocatalyst is characterized by its ease of synthesis, low cost, nontoxicity, high stability, high band-gap energy (3.2 eV) [15], and applicability for PCD [16]. Despite these qualities, it is now of little use in large-scale processing operations due to its low quantum efficiency and short wavelength excitation. According to the concept of irradiation, when the particles are irradiated with near ultraviolet rays of energy greater than or equal to 350 nm [17], the electrons migrate from the valence band to reach the conduction band (e_cb_^−^) leaving behind positive holes (h_vb_^+^), where they initiate oxidation reactions and reduction reactions with the absorbed species [18]. Such charges can be recombined to generate an e^−^-hole pair that is not available to participate in the required reactions. Before this process takes place, oxidation and reduction processes occur by means of electrons and holes, which control the efficiency of the photocatalytic TiO_2_-catalyzed reaction [19,20]. 

One of the approaches that has been widely adopted to improve the photocatalytic efficiency of the TiO_2_ photocatalyst under UV and visible light is the doping/deposition of a minute amount of a suitable metal in TiO_2_, which enhances the participation of the photogenerated holes and electrons in the photocatalytic reactions [21,22,23,24,25]. It was suggested that the precipitated or doped metal in TiO_2_ has high Schottky barriers in the metal–TiO_2_ contact area and thus acts as an electronic trap, which facilitates the separation of the electron hole and enhances the electron transfer process of the interface, thus enhancing the photocatalyst’s efficiency [26]. According to the higher energy gap of TiO_2_ NPs, one way to make these particles useful in terms of photocatalytic activity is by doping the TiO_2_ NPs with rare earth ions, where these ions decrease the metal oxide band gap [27]. Visible light represents about 43% of the solar spectrum, so researchers have aimed to improve the absorption of visible light by TiO_2_ due to its previously mentioned properties and applications in the visible and ultraviolet frequency ranges [28,29,30]. 

This article attempts to explain and develop the existence of erbium ions with optimum quantities and their role in the physicochemical properties of the doped TiO_2_ materials, especially when used as a coating filter. The best optical characteristics were obtained by the prepared nanocomposites by designing a coating layer on a contact lens front surface to fabricate a blue-blocking contact lens. In the continuation of our work [31,32,33,34,35] on nanomaterials, herein, we report the synthesis of TiO_2_ NPs and Er^3+^/TiO_2_ by the impregnation method [36,37,38].

## 2. Materials and Methods

The chemicals erbium (III), TiO_2_ NPs powder, and chloroxylenol were purchased from Sigma Aldrich (St. Louis, MO, USA) and utilized without further purifications. The Er^3+^ contained acetylacetonate hydrate, was CAS grade, and had a purity of 97%. The TiO_2_ nanopowder had 99.5% purity and a 21 nm diameter according to the TEM from Philips (Eindhoven, The Netherlands), and the chloroxylenol had 98% purity. The chloroxylenol was prepared, and deionized water was used for this purpose as well as for dilution.

### 2.1. Er^3+^/TiO_2_ Preparation

TiO_2_ NPs was impregnated with Er^3+^ ions with 0.6, 1.2, 1.8, and 2.4 wt % using an 80 mL glass beaker. We used a 10 mL of deionized water to dissolve 60, 120, 180, and 240 mg of Er^3+^ acetylacetonate hydrate, which was sonicated for 10 min. Then, each solution was added dropwise to the TiO_2_ nanopowder, well-mixed with a glass rod, and stirred for 2 h at 1300 rpm. The drying process was carried out at 80 °C overnight and then calcination was performed at 500 °C for two hours.

### 2.2. Er^3+^/TiO_2_ Characterization

The estimation of Er^3+^/TiO_2_ NPs phase and size was achieved using D8 Advance Bruker AXS X-ray patterns (Bruker, Am Studio 2D, Berlin, Germany). Energy-dispersive X-ray (EDX), with higher resolution, was applied to investigate the synthesized NPs’ morphological features using field emission scanning electron microscopy (FESEM) (SuPRA 55 VP, Carl Zeiss AG, Oberkochen, Germany). The other two analysis devices were used to characterize the NP samples were an OXFORD Penta FETx3 for metal mapping (Carl Zesis AG, Oberkochen, Germany) and a Perkin Elmer UV–Vis spectrophotometer (Waltham, MA, USA). The intensity transmission curve of the designed coated contact lens was recorded by ZEMAX software.

### 2.3. Contact Lens Modeling

To design and fabricate an anti-contamination blue wavelength blocker contact lens, the UV–Vis spectral data for all the synthesized films were compared; the best *n* value nanocomposite was selected for coating the contact lens. The human eye optical parameters and the coated contact lens at a 5 degrees field of view under white light illumination were entered into ZEMAX software (Version 14, serial number 34,900). The two corneal surfaces (anterior and posterior cornea) were selected as aspherical surfaces (programmatically named the standard surface). The standard surface requires two specific parameters: radius and conic constant. The pupil aperture surface was selected as the stop aperture and nasally decentered by 0.5 mm with respect to the visual axis; the crystalline lens is represented by two homogeneous gradient index shells whose *n* value is described by:(1)$$n=n_{0}+n_{r2}r^{2}+n_{r4}r^{4}+n_{r6}r^{6}+n_{z1}z+n_{z2}z^{2}+n_{z3}z^{3}$$
(2)$$r^{2}=x^{2}+y^{2}$$

Both the vitreous body of the eye and the retinal imaging surface were selected as standard surfaces. The standard surface position is centered on the optical axis and its vertex located at the *Z* axis. The *z* value (sag) of the standard surface is given by:(3)$$z=\frac{c\;r^{2}}{1+\sqrt{1-(1+k\;c^{2}\;r^{2})}}$$
where *c* is the curvature (reciprocal of the radius), *r* is the radial coordinate in the lens unit, and *k* refers to the conic constant. 

In this work, a coating of aspheric surface of the modeled contact lens (0.12 mm thickness, 0.035 conic, and 7.748 mm and 7.8 mm radius of front and back surfaces, respectively) was constructed by an extended polynomial surface. The aspherical surface sag is defined by: (4)$$z=\frac{c\;r^{2}}{1+\sqrt{1-(1+k)\;c^{2}\;r^{2}}}+\sum_{i=1}^{N}A_{i}E_{i}(x,y)$$
where *N* represents the polynomial coefficient number in the series and *A_i_* is the coefficient on the *i*th extended polynomial (*E*) term

## 3. Results

### 3.1. X-ray Diffraction Pattern

The synthesized specimens were analyzed by XRD in order to study the crystallinity of the erbium ions/titanium dioxide (Er^3+^/TiO_2_) NPs and to evaluate the changes, if any, that may occur through the impregnation of TiO_2_ NPs with *Er* ions. The X-ray diffraction scanning of the Er^3+^/TiO_2_ NPs is demonstrated in Figure 1. The signals of the TiO_2_ types, which are anatase and rutile [37,38], were clear. These patterns demonstrate that the phase composition and the structure of the TiO_2_ NPs did not change even after impregnation with *Er* ions. Additionally, the decrease in the intensity of the TiO_2_’s peaks indicated that the crystal structure of the titanium dioxide was impregnated with *Er* ions.

Table 1 demonstrates the intensity values of the signals. The diffraction spectra of the *Er* ions/TiO_2_ NPs do not support the existence of *Er* ions. This conclusion can most probably be ascribed to the appropriate dispersion of erbium ions in the lattice structure of TiO_2_ [39].

The crystallite size data are listed in Table 2 for TiO_2_ with *Er* ions in weight percent [15]. The lesser increase in the size of the particles confirmed that impregnation increased for the *Er* ions/TiO_2_ NPs sample. The Debye–Scherer equation is used to calculate the crystallite size *D* (Equation (5)):(5)$$D=\frac{K\lambda }{\beta \cos\theta }$$
where the Scherer constant (K) is equalto 0.89, *λ* is the wavelength, *β* is the width of the peak, and *θ* is the angle of Bragg diffraction.

### 3.2. Surface Morphology

The Er^3+^/TiO_2_ NPs surface morphologies were examined utilizing FESEM and EDX in addition to metal mapping. The FESEM images are demonstrated in Figure 2. The images of the Er^3+^/TiO_2_ sample demonstrate the particles’ shapes, which showed less agglomeration and were generally spherical. EDX analysis was performed on the Er^3+^/TiO_2_ samples to study the presence of *Er* ions on the surfaces of the examined TiO_2_. The results of EDX spectra analysis are provided in Figure 3, which demonstrates that the *Er* ions were present in all investigated samples. In addition, the EDX analysis spectral results of the different weight percentages of the *Er* content in the tested samples are listed in Table 3. The theoretical and empirical *Er* contents show considerable agreement. Metal mapping tests were performed to study the *Er* ions’ dispersion on the surface of the TiO_2_ NPs. These data are summarized in Figure 4, which reveals a considerable distribution of *Er* ions on the TiO_2_ NPs in all synthesized samples.

### 3.3. Energy Gaps

The absorbance spectra of the synthesized NPs samples were obtained at room temperature and are illustrated in Figure 5. A significant absorption peak in the UV–Vis spectrum is located at 320 nm. According to Equation (6), the energy gaps for the synthesized samples of NPs were calculated: (6)$$\alpha h\nu =E_{d}{\left(h\nu -E_{g}\right)}^{\frac{1}{2}}$$
where *α* represents the optical coefficient; *h* refers to the Planck constant; the multiplication of *hv* represents the photon energy; *Eg* and *E_d_* represent the energy gap and a constant, respectively [40]. (*αhv*) versus *hv* was drawn, the linear section of the curve was extrapolated, and Eg was evaluated, as shown in Figure 6. The *Eg* values of TiO_2_ NPs for different weight percentages of *Er* ions are given in Table 4. The lowest energy gap value obtained was 2.63 eV for *Er* ions/titanium dioxide NPs.

### 3.4. Application as a Blue-Wavelength-Blocking Contact Lens

Given most people’s modern lifestyles, they spend the majority of their waking hours staring at digital devices. According to the previous studies, 60% of people spend 6–7 h per day looking directly at digital screens [41,42]. Digital screens, sunlight, electronics, and fluorescent light sources emit a blue wavelength ranging from 300 to 500 nm. This range is harmful to the eye (damages the retinal photoreceptors and destroys the crystalline lens protein that appears as a white cloud) [43]. 

The human eye’s natural filters do not protect the eye’s components against these highest-energy wavelengths. The excessive exposure to blue light rays may affect the retinal tissues and cause damage. This contributes to age-related macular degradation and subsequent loss of vision. In this work, 0.6% wt/vol Er^3+^/TiO_2_ solution drops were spin-coated on a previously prepared poly(methyl methacrylate)—titanium dioxide (PMMA-TiO_2_) contact lens fixed on a convex glass mold (Figure 7a) [11]. The refractive index was measured using an Abbe Refractometer at wavelengths of 486.1, 587.6, and 656.3 nm as 1.509, 1.503, and 1.499, respectively. The Abbe number was calculated as 52.35 using the following equation [44]:(7)$$\nu_{d}=\frac{\left(n_{d}-1\right)}{\left(n_{f}-n_{c}\right)}$$
where *ν_d_* is the Abbe number; and *n_c_*, *n_d_*, and *n_f_* are refractive indices of the polymer films at wavelengths of 656, 589, and 486 nm, respectively.

From the results provided above, it can be found that the fabricated coated contact lens works in the acceptable physical range according to the refractive index, which is near the refractive indices of the eye tissues. The resulting contact lens was modeled using ZEMAX optical design software (version 14) to evaluate the blue-wavelength-blocking characteristics before conducting an in vivo study. The Liou and Brennan eye model was chosen in this study because it was experimentally obtained from the human eye [45]. The Er^3+^/TiO_2_ coating worked as a blue wavelength filter, where it prevented the wavelength range of 0.45–0.47 nm from being transmitted (reduced the light transmittance to 0%) through the contact lens, as illustrated in Figure 7c. 

The Er^3+^/TiO_2_ coating does not degrade the retinal image’s clarity, as shown in Figure 8. To compare the retinal image before and after coating, image simulation was used for evaluation. A polychromatic light source was applied to the modeled eye, a perfect eye with a high retinal image contrast, and sharpness was obtained by the eye with the PMMA-TiO_2_ contact lens with no aberrations, as shown in Figure 8a. After, the resultant PMMA-TiO_2_ contact lens coated with Er^3+^/TiO_2_ layer was fitted on the modeled eye, and the image at the retina showed high clarity and no indication of aberration generation. This indicates that the coated lens focused the rays entering the eye in one small spot, which is the focal point. This performance was obtained due to *n* value of the resulting lens and the dispersion factor, which is able to maintain image clarity under polychromatic light illumination. The speed of each component of white light differs from the speed of the other wavelengths according to the concept of the refractive index (*n* = c/v). The blocking lens color is suitable and not too deep, so it prevents color deviation and darkness. The designed contact lens can work as a filter or polarizer, blocking the reflected blue portion of light from the object and passing the other components of the light through to the eye. From observing the sharpness and contrast of the two images before and after the application of the coating, we found the coating does not affect the visual function of the eye.

## 4. Conclusions

Titanium dioxide NPs were impregnated with different weight percentages of *Er* ions using the impregnation technique and then calcination of the product at 500 °C. The Er^3+^/TiO_2_ NPs showed an energy gap decrease with increasing *Er* ions weights percent up to 1.8 wt %. Agglomeration and particle size increased with increasing *Er* ions weight percent. Relative to the long hours people spend staring at digital displays, researchers are developing eyeglass lenses that partially filter the high-energy blue wavelength. Due to the problems with glasses such as their weight, the possibility of breaking them, amongst others, we attempted to design and manufacture a contact lens with a high *n* value that works as a filter to reduce the transmission of blue light intensity without affecting the clarity of the image. The fabricated spin-coated Er^3+^/TiO_2_ layer on the PMMA-TiO_2_ contact lens works as a filter as it absorbs the high-energy blue wavelength and passes other components of light through to the eye. A perfect polarized retinal image was achieved, and no aberration was generated.

## Figures and Tables

**Figure 1 nanomaterials-11-02190-f001:**
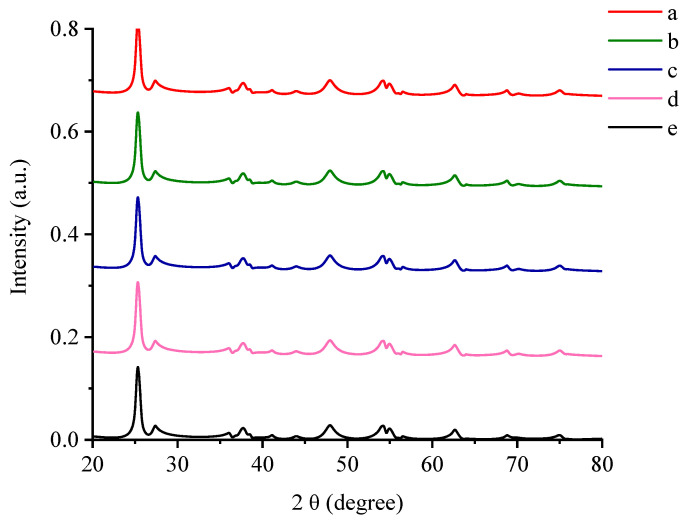
XRD: (**a**) titanium dioxide; (**b**) 0.6 wt %; (**c**) 1.2 wt %; (**d**) 1.8 wt %; (**e**) 2.4 wt % erbium ions/titanium dioxide.

**Figure 2 nanomaterials-11-02190-f002:**
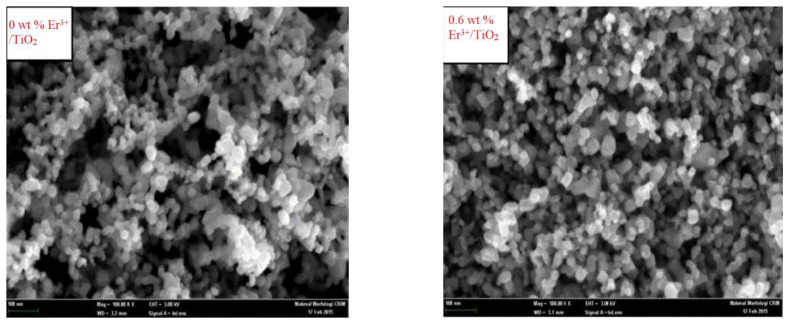
FESEM micrographs for the synthesized samples.

**Figure 3 nanomaterials-11-02190-f003:**
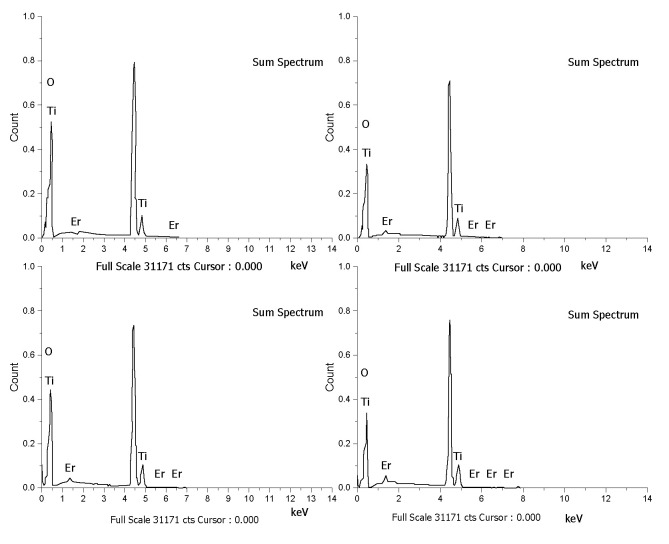
EDX micrographs of the synthesized samples.

**Figure 4 nanomaterials-11-02190-f004:**
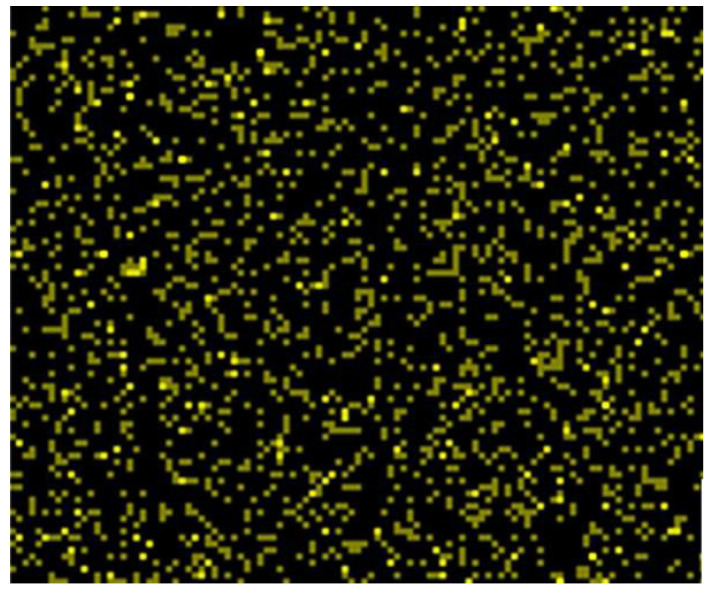
Metal mapping of the synthesized Er^3+^/TiO_2_ NPs.

**Figure 5 nanomaterials-11-02190-f005:**
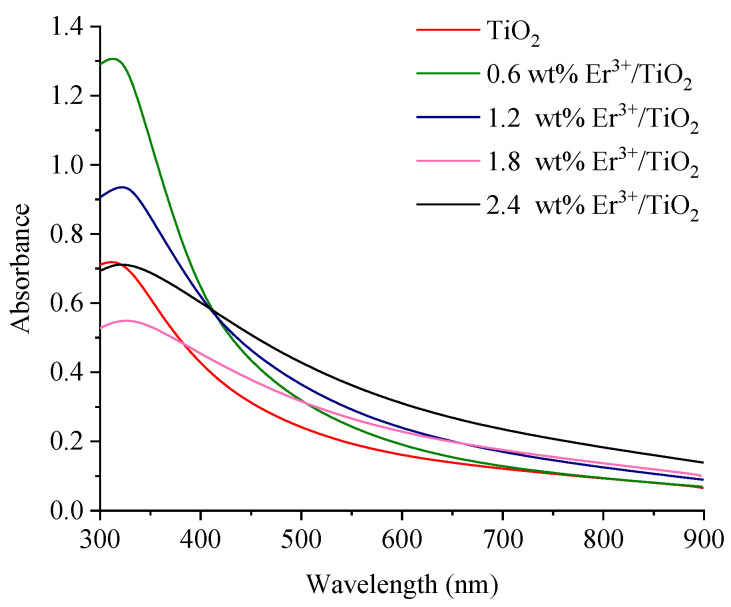
Ultraviolet–visible spectra of the synthesized NPs.

**Figure 6 nanomaterials-11-02190-f006:**
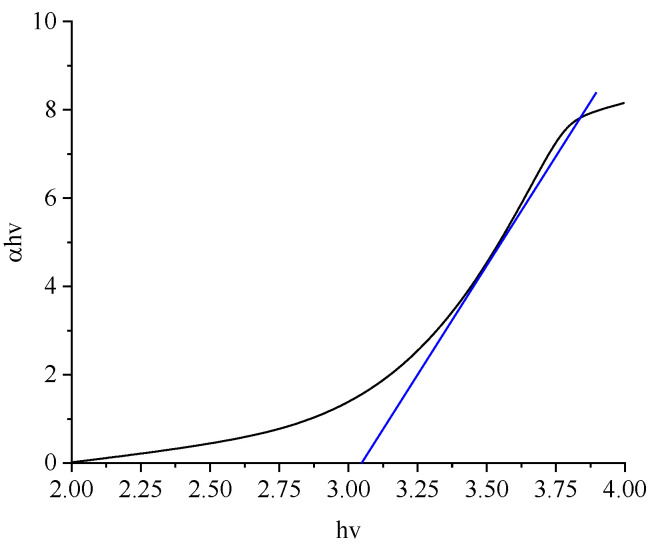
Calculation of the direct band gap energy obtained in our study.

**Figure 7 nanomaterials-11-02190-f007:**
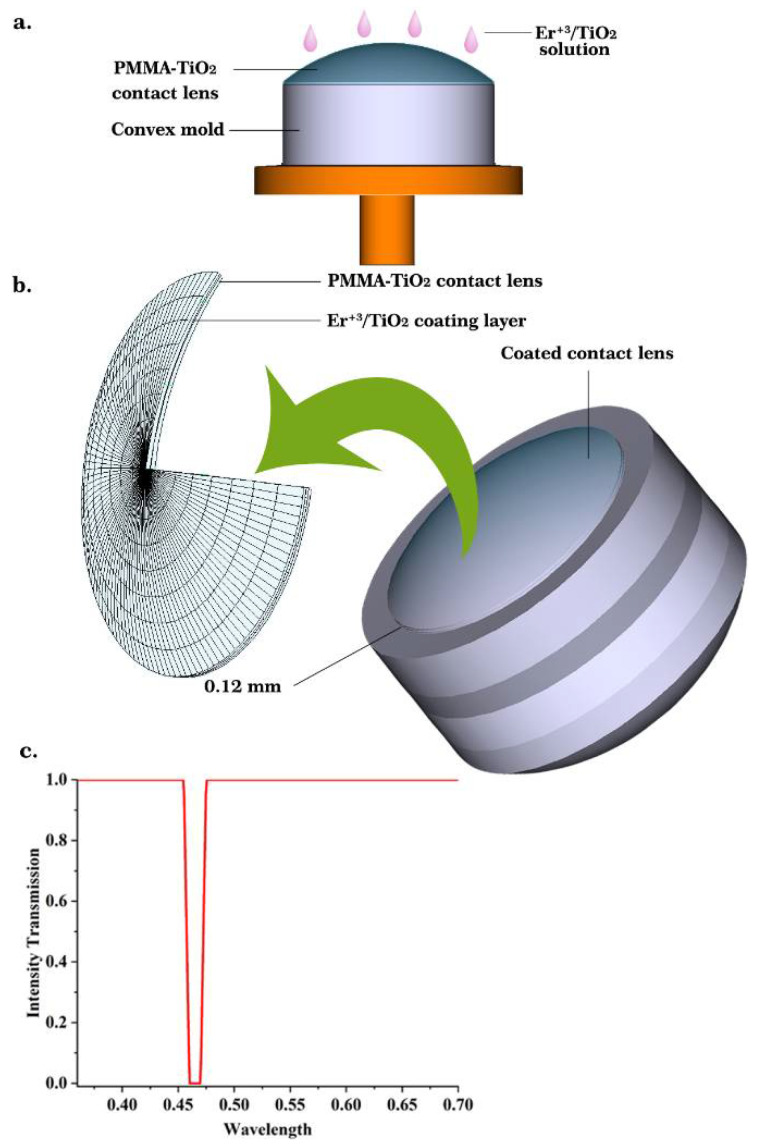
(**a**) Spin-coating a contact lens with 0.6 wt/vol % Er^3+^/TiO_2_ solution; (**b**) The layers of the resulting contact lens; (**c**) intensity transmission as a function of wavelength blocking 0.45–0.47 nm from transmission.

**Figure 8 nanomaterials-11-02190-f008:**
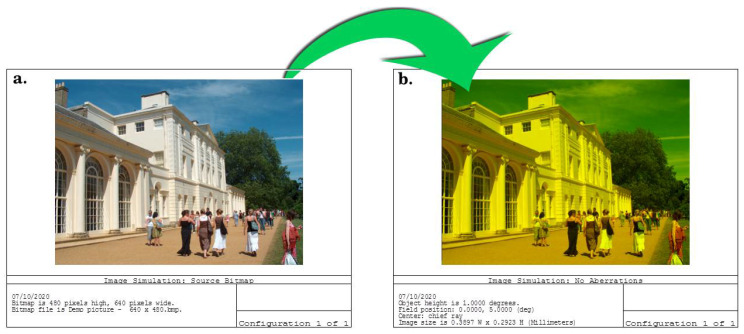
(**a**) The retinal image resulting from the eye with a PMMA-TiO_2_ contact lens; (**b**) the retinal image resulting from the eye with a PMMA-TiO_2_ contact lens coated with Er^3+^/TiO_2_ NPs.

**Table 1 nanomaterials-11-02190-t001:** Erbium ions/titanium dioxide intensity signals.

2θ	Intensity				
	TiO_2_	0.6 wt % Er^3+^/TiO_2_	1.2 wt % Er^3+^/TiO_2_	1.8 wt % Er^3+^/TiO_2_	2.4 wt % Er^3+^/TiO_2_
25.2712	2158	1984	1911	1879	1855
27.4136	515	498	493	481	474
37.3038	127	105	100	90	90
48.0908	555	536	481	478	473
53.8206	311	310	288	262	230
53.97	379	368	351	346	336
55.116	367	340	313	291	283
62.7142	292	285	272	249	241
68.8675	165	149	143	131	120
70.2128	145	114	109	108	104
75.0956	185	164	163	151	147

**Table 2 nanomaterials-11-02190-t002:** Size of crystallite particles of all synthesized samples.

Er^3+^/TiO_2_ (wt %)	Crystallite (nm)
TiO_2_	28.72
0.6	28.84
1.2	31.53
1.8	33.93
2.4	35.93

**Table 3 nanomaterials-11-02190-t003:** EDX analysis of the theoretical and empirical values of the synthesized samples.

Er^3+^/TiO_2_ (wt %)	Calculated Metal (wt %)	Actual Metal (wt %) by EDX
0.6	0.6	0.72
1.2	1.2	1.31
1.8	1.8	2.06
2.4	2.4	2.45

**Table 4 nanomaterials-11-02190-t004:** Energy gaps for the synthesized Er^3+^/TiO_2_ NP samples.

Er^3+^/TiO_2_ (wt %)	Energy Gap (eV)
TiO_2_	3.13 ± 0.02
0.6	3.20 ± 0.02
1.2	3.00 ± 0.02
1.8	2.63 ± 0.02
2.4	2.70 ± 0.02

## Data Availability

The data of this study are available from the corresponding author (A.A.), upon reasonable request.

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
