# Peer review of "Novel Blue-Wavelength-Blocking Contact Lens with Er3+/TiO2 NPs: Manufacture and Characterization"

_nanomaterials, 2021, doi:10.3390/nano11092190_

Round 1

Reviewer 1 Report

This is a good article that might be suitable for publication after taking into account the following suggestion, which will certainly improve this manuscript.

In the first long paragraph of Introduction, which is very informative, there are no supporting links that would be helpful to readers. Please add some several important references in the end of paragraph.

For Eg of TiO2 please provide reference. Probably, it is also important to for which structure this value is valid.

Please note somewhere in the text that besides the "TiO2 synthesis impregnation method", other methods are also being developed. See few recent MDPI papers: Crystals 2021, 11(4), 431; https://doi.org/10.3390/cryst11040431 Crystals 2021, 11(7), 794; https://doi.org/10.3390/cryst11070794

Could you discuss in which sites of the TiO2 lattice are Er ions located - close to the surface or in the bulk? Is there a tendency for them to agglomerate?

Have the test studies of the stability of the considered systems from time of use and from ambient temperature been successful? Have any degradation effects been identified?

Author Response

Dear reviewer,

Thank you for your useful comments and suggestions, all have been done

Please see the attached file and the revised manuscript

Best regards

Reviewer 2 Report

This is actually a good piece of work that is undoubtedly good enough to get published, but after introducing some important improvement of the current manuscript.

  1. In the first long paragraph  of  the Intruduction  there is no any supporting references. Please provide  few important sources for the line 37.
  2. line 41, please not that  even nanoparticles of perovskites ( for example of BaZrO3) can be obtained: https://aip.scitation.org/doi/abs/10.1063/1.4959020 https://www.tandfonline.com/doi/abs/10.1080/15421406.2013.873646
  3. Line 45. For Eg of TiO2 please provide reference.
  4. Line 65.  "... to decrease metal band gap".  Did you mean "metal oxide"?
  5. Line  75 please compare "impregnation method of the synthesis TiO2" with others, for example,  recently developed, extraction-pyrolytic method (Eu:TiO2) https://doi.org/10.1016/j.jmrt.2021.06.029
  6.  In Fig.4 please add the vertical axis 
  7. Could you discuss the preferable lattice sites of Er TiO2 - close to the surface ot in the bulk.

Author Response

(The authors gave the same response as above.)
